# Bibliometric Analysis and Visualization Mapping of Anthrax Vaccine Publications from 1991 through 2021

**DOI:** 10.3390/vaccines10071007

**Published:** 2022-06-23

**Authors:** Tauseef Ahmad, Mukhtiar Baig, Sahar Shafik Othman, Husam Malibary, Shabir Ahmad, Syed Majid Rasheed, Mohammad T. Al Bataineh, Basem Al-Omari

**Affiliations:** 1Vanke School of Public Health, Tsinghua University, Beijing 100084, China; 2Department of Epidemiology and Health Statistics, School of Public Health, Southeast University, Nanjing 210096, China; 3Department of Clinical Biochemistry, Faculty of Medicine, Rabigh, King Abdulaziz University, Jeddah 25289, Saudi Arabia; mbbaig@kau.edu.sa; 4Department of Family Medicine, Faculty of Medicine, King Abdulaziz University, Jeddah 25289, Saudi Arabia; sahar.s.othman@gmail.com; 5Department of Internal Medicine, Faculty of Medicine, Rabigh, King Abdulaziz University, Jeddah 25289, Saudi Arabia; hmmalibary@kau.edu.sa; 6Department of Agriculture, Bacha Khan University Charsadda, P.O. Box 20, Charsadda 24420, Khyber Pakhtunkhwa, Pakistan; shabiramp99@gmail.com (S.A.); smrasheed@bkuc.edu.pk (S.M.R.); 7Center for Biotechnology, Khalifa University of Science and Technology, Abu Dhabi P.O. Box 127788, United Arab Emirates; mohammad.bataineh@ku.ac.ae; 8Emirates Bio-Research Center, Ministry of Interior, Abu Dhabi P.O. Box 127788, United Arab Emirates; 9Department of Epidemiology and Population Health, College of Medicine and Health Sciences, Khalifa University, Abu Dhabi P.O. Box 127788, United Arab Emirates; 10K.U. Research and Data Intelligence Support Center (RDISC) AW 8474000331, Khalifa University of Science and Technology, Abu Dhabi P.O. Box 127788, United Arab Emirates

**Keywords:** anthrax, vaccine, bibliometric analysis, visualization mapping

## Abstract

Purpose: This study aims to analyze and characterize anthrax vaccine-related research, key developments, global research trends, and mapping of published scientific research articles during the last three decades (1991–2021). Methods: A bibliometric and visualized study was conducted. The Web of Science Core Collection database (WoSCC) was searched using relevant keywords (“Anthrax” OR “Anthrax bacterium” OR “*Bacillus anthracis*” OR “Bacteridium anthracis” OR “Bacillus cereus var. Anthracis” (Topic)) AND (“Vaccine” OR “Vaccines” OR “Immunization” OR “Immunisation” OR “Immunizations” OR “Immunisations” (Topic)) with specific restrictions. The data was analyzed and plotted by using different bibliometric software and tools (HistCiteTM software, version 12.3.17, Bibliometrix: An R-tool version 3.2.1, and VOSviewer software, version 1.6.17). Results: The initial search yielded 1750 documents. After screening the titles and abstracts of the published studies, a total of 1090 articles published from 1991 to 2021 were included in the final analysis. These articles were published in 334 journals and were authored by 4567 authors from 64 countries with a collaboration index of 4.32. The annual scientific production growth rate was found to be 9.68%. The analyzed articles were cited 31335 times. The most productive year was 2006 (*n* = 77, 7.06%), while the most cited year was 2007 (2561 citations). The leading authors and journals in anthrax research were Rakesh Bhatnagar from Jawaharlal Nehru University, India (*n* = 35, 3.21%), and *Vaccine* (*n* = 1830, 16.51%), while the most cited author and journal were Arthur M. Friedlander from the United States Department of Defense (*n* = 2762), and *Vaccine* (*n* = 5696), respectively. The most studied recent research trend topics were lethal, double-blind, epidemiology, B surface antigen, disease, and toxin. The United States of America (USA) was the most dominant country in terms of publications, citations, corresponding author country, and global collaboration in anthrax vaccine research. The USA had the strongest collaboration with the United Kingdom (UK), China, Canada, Germany, and France. Conclusion: This is the first bibliometric study that provides a comprehensive historical overview of scientific studies. From 2006 to 2008, more than 20% of the total articles were published; however, a decrease was observed since 2013 in anthrax vaccine research. The developed countries made significant contributions to anthrax vaccine-related research, especially the USA. Among the top 10 leading authors, six authors are from the USA. The majority of the top leading institutions are also from the USA. About 90% of the total studies were funded by the United States Department of Health and Human Services (HHS), National Institutes of Health (NIH), USA, and the National Institute of Allergy and Infectious Diseases (NIAID), USA.

## 1. Introduction

For centuries, humans have been plagued by outbreaks of deadly zoonotic diseases with epidemic potential. *Bacillus anthracis*, a Gram-positive, aerobic, endospore-forming rod belonging to the genus Bacillus, is the causative agent of anthrax, a zoonotic disease [1,2,3]. However, prevalence data is scarce from many parts of the developing world about endemic diseases [4]. As a result, anthrax is a neglected disease with a hazy global distribution [5]. More than a billion people reside in anthrax-prone areas, most of whom live in African, European, and Asian rural regions [5]. However, most of that population is unlikely to have had any occupational exposure to infected animals, and direct soil exposure has only been reported in a few human cases [5].

Animals or animal products almost always cause humans to contract a natural disease, either directly or indirectly [2]. For example, more than 95 percent of human cases of anthrax are caused by cutaneous anthrax, which can be effectively treated with antibiotics [6]. Furthermore, inhalation of anthrax has a 90 percent mortality rate if untreated and can be used as a biological weapon [7].

In anthrax infections, two sets of genes are largely responsible for pathology and virulence. The pXO1 plasmid contains “the protective antigen (PA), lethal factor (LF), and edema factor (EF)” genes. Lethal toxin (LT) and edema toxin (ET) are formed when PA binds to LF and EF, respectively. Binding by PA allows LF and EF to enter cells, causing cellular toxicity and contributing to the disease’s lethality [8,9,10,11,12]. The pXO2 plasmid contains genes for capsule production and regulation and is also involved in anthrax disease [13]. Anti-PA and toxin-neutralizing antibodies are critical components of safety due to PA’s fundamental role in the toxic outcomes of anthrax infections, and PA is an essential antigen in current vaccine development [14].

*Bacillus anthracis* is an extremely virulent bacterium whose spores can survive for long periods in the environment, are easily transmitted, and are linked to high morbidity and mortality rates. As a result of these factors, anthrax has received increased consideration as a prospective agent for bioterrorism and warfare; the authorities are concerned with building up stocks of anthrax vaccines to protect public health by employing mass immunization if needed [15,16].

The first anthrax vaccine for human use was produced in the 1950s in the United States of America (USA), and in 1970 it was permitted for usage [14,17,18]. However, until the 2001 Amerithrax attacks, no further advancement was made regarding its use as a “post-exposure prophylaxis” [14]. Following the 2001 attack, new investments in anthrax medical defensive measures were made. Since then, significant advancement has been made with both the permitted vaccine and next-generation vaccine candidates [14].

In 1970, AVA or Anthrax vaccine adsorbed (BioThrax^®^) was permitted for pre-exposure prophylaxis. BioThrax^®^ is made from cell-free extracts of an avirulent strain of B. anthracis [19]. It can produce antibodies against the P.A. protein, neutralizing anthrax toxins in a nonclinical anthrax challenge model [19]. The current BioThrax^®^ pre-exposure prophylaxis schedule includes a three-dose primary series of intramuscular injections (at 0, 1, and 6 months), followed by booster vaccinations at 6 and 12 months. Following that, annual boosts are needed [20]. Hopkins and colleagues published a Phase 3 study showing the probable effectiveness of post-exposure prophylaxis when three doses were administered subcutaneously (at 0, 2, and 4 weeks) [21].

In the United Kingdom (UK), anthrax vaccine precipitated (AVP) is approved for use. AVP is an alum-precipitated cell-free filtrate of the B. anthracis Sterne 34F2 strain. AVP incorporates the 3 key anthrax toxin elements, i.e., P.A., L.F., E.F. [22]. AVP, like the BioThrax^®^ pre-exposure prophylaxis regimen, is administered intramuscularly. Doses are administered at 0, 3, 6, and 32 weeks, with annual boosts [23].

A live-attenuated anthrax vaccine is permitted for human cutaneous and subcutaneous administration in Russia. In the 1940s, this vaccine was first established, and two distinct nonencapsulated B. anthracis variants’ live dry spores were used in this vaccine [24]. One of the strains is now combined with P.A. adsorbed on aluminum hydroxide in the current formulation. This formulation requires three years of annual subcutaneous injections and two-year boosters [24]. In addition, a live attenuated anthrax vaccine in the form of a suspension of the attenuated strain A16R is also available for human use in China [25].

AV7909 (NuThrax TM) is a next-generation anthrax vaccine that comprises the previously permitted BioThrax^®^ adjuvanted with the TLR9 agonist CPG 7909. CPG 7909 is an oligodeoxynucleotide that has been shown to augment vaccine immunogenicity by activating B-cells [26]. When AV7909 was compared to BioThrax^®^, applying a 2-week vaccination schedule (0 and 14 days), clinical studies revealed that AV7909 yielded anthrax toxin neutralizing antibody titers faster than BioThrax^®^ and had a substantially greater response [27]. In addition, a Phase 2 clinical trial with a two-dose regimen on days 0 and 14 was added to the immunogenicity data, bolstering the case for AV7909’s use [28].

To provide vaccines that are easier to manufacture and administer, a variety of vaccines based on recombinant P.A. (rPA) (produced in plants or bacterial production platforms) have been developed [14]. Two more vaccine contenders established on P.A. have tried to deliver the antigen using viral vectors. P.A. was expressed using Adenovirus serotype 4 and was tested in phase 1 clinical trial. AdVAV, a replication-deficient adenovirus type 5 vectored P.A., was recently tested in a rabbit challenge model and found to protect against lethal aerosol exposure [29]; since then, it has advanced to clinical trials. Intranasal exposure would be used to administer AdVAV. Many funding agencies and sponsors have pursued efforts to improve the vaccine schedule and dosing strategies. In 2015, the USA approved a vaccine permitted for post-exposure prophylaxis when combined with antibiotics [20]. Thus, the current study was conducted to highlight the global research output and trends, key developments, and network visualization mapping of articles published on the anthrax vaccine between January 1991 and December 2021. Bibliometric type studies are widely used in different research areas to examine the knowledge structure, research output, achievements, and developments [30,31].

## 2. Methods

### 2.1. Study Design and Searching Strategy

A bibliometric-visualized study was conducted utilizing the Web of Science Core Collection (WoSCC) database hosted by Clarivate Analytics. In the WoSCC, the search was limited to publication year (1904–2021), document type (article), publishing language (English), and WoSCC index (Science Citation Index Expanded (SCI-Expanded)). The WoSCC database was preferred because it is widely used in bibliometric studies and provides essential information on journals and other bibliometric indices [32,33,34,35,36,37]. The online search was performed on 13 October 2021 and updated on 27 May 2022, by the lead author (TA).

The potential keywords were extracted from the published literature and searched with the topic field. The topic field searches for title, abstract, author keywords, and KeyWords Plus. The following search query was adopted with specific restrictions “Anthrax” OR “Anthrax bacterium” OR “*Bacillus anthracis*” OR “B. anthracis” OR “Bacteridium anthracis” OR “Bacillus cereus var. Anthracis” (Topic) and “Vaccine” OR “Vaccines” OR “Immunization” OR “Immunisation” OR “Immunizations” OR “Immunisations” (Topic).

### 2.2. Data Downloading and Extraction

The complete records of the retrieved articles were downloaded or manually entered into a Microsoft Excel spreadsheet (Microsoft Corporation, Redmond, WA, USA), and any discrepancy or disagreement was discussed and resolved with other authors. The dataset was downloaded both in a Tab-delimited file and a Comma-separated values format. The included studies were organized in descending order based total number of citations. Several attributes were extracted from the selected articles, including authors’ names, journals, publications year, most studied research areas, frequently used keywords, citations count, and country or region of origin. The journals’ Impact Factor (IF) and Quartile ranking (Q1–Q4) for the year 2020 were obtained from Incites Journal Citation Reports (released in June 2021 by Clarivate Analytics, London, UK).

### 2.3. Data Analysis and Interpretation

The obtained data was exported into the Microsoft Excel 2019, HistCiteTM software version 12.3.17, RStudio (Bibliometrix: An R-tool version 3.2.1), and VOSviewer software version 1.6.17 for windows. The calculated values were presented in frequencies (*n*) and percentages (%). The required graphs were generated using Microsoft Excel 2019. Then, the data was exported into VOSviewer for network visualization. The VOSviewer is a freely available and commonly used tool for visualization mapping [38]. The citation frequency was calculated using HistCite^TM^ for windows [39]. Furthermore, the data was plotted for co-authorship countries and co-occurrence keywords visualization mapping using VOSviewer software. In addition, the data was exported into RStudio for trend analysis and thematic mapping.

## 3. Results

The initial search retrieved 1750 documents. From those, 524 documents were excluded for being published in a form other than an article, 19 were not published in English, and 69 were not indexed in SCI-Expanded, leaving 1138 articles. The number of articles published between 1904 and 1990 was limited in number and scattered over a long period (*n* = 34 over 86 years). Therefore, the final analysis included articles published between 1 January 1991 and 31 December 2021 (*n* = 1090). Figure 1 shows the articles’ selection flow chart. The included articles were published in 334 journals and were authored by 4567 authors (4.19 authors per article). A total of 1151 institutions from 63 countries participated in anthrax vaccine-related research. These articles were cited 6327 times locally and 31,335 times globally. The overall collaboration index among authors was 4.32. The annual scientific production growth rate was 9.68%. Table 1 shows the main description of the included articles.

The citations for the top 10 articles range from 215 to 633 citations in the WoSCC database. The highest cited article was “The genome sequence of *Bacillus anthracis* Ames and comparison to closely related bacteria”, published in *Nature* (2003), which received 633 total citations (33.32 citations per year) [40]. Of the total 1090 articles, 54 (5%) articles did not receive any citation and 35 (3.2%) articles received only 1 citation each. Two articles were cited more than 500 times, and ten articles were cited more than 200 times.

The most productive years in terms of published articles in anthrax vaccine research were 2006, 2007, and 2008, (*n* = 77, 7.06%), (*n* = 73, 6.70%), and (*n* = 71, 6.51%), respectively, as shown in Figure 2. The R^2^ value between the number of articles and publication year was calculated using a linear regression model. The R^2^ value was found to be 0.35. The red line in Figure 2 represents the linear trend line. The most cited year was 2007 (*n* = 2561), followed by 2006 (*n* = 2553), and 2003 (*n* = 2413), as shown in Table 2.

### 3.1. Highest Publishing Authors and Journals

The author who published the highest number of articles related to the anthrax vaccine was Rakesh Bhatnagar from Jawaharlal Nehru University, India (*n* = 35, 3.21%), followed by Stephen H. Leppla from the National Institutes of Allergy and Infectious Diseases (NIAID), USA (*n* = 34, 3.2%), then Conrad P. Quinn from Centers for Disease Control and Prevention (CDC), USA (*n* = 31, 2.84%). The most cited authors were Arthur M. Friedlander (*n* = 2762), Stephen H. Leppa (*n* = 1731), and Bruce E. Ivins (*n* = 1665). Among the top 10 leading authors, six authors are from the USA. Based on the articles’ fractionalized frequency the top contributing authors were Rakesh Bhatnagar (8.72), Stephen H. Leppla (6.36), and Stephen F Little (5.42). The fractionalized frequency quantifies an individual author’s contribution to a published set of papers. Table 3 shows the details of authors who published at least 15 anthrax vaccine-related articles.

The journal that published the most anthrax vaccine-related articles was *Vaccine* (*n* = 180, 16.51%), followed by *Infection and Immunity* (*n* = 103, 9.45%), then *Clinical and Vaccine Immunology* (*n* = 42, 3.85%). *Vaccine* is hosted by Elsevier, while the other two journals are published by the American Society for Microbiology. Furthermore, the most cited journals were *Vaccine* (*n* = 5695), *Infection and Immunity* (*n* = 5333), and *Proceedings of the National Academy of Sciences (PNAS) of the USA* (*n* = 2008). Table 4 shows the details of journals that published at least 15 anthrax vaccine-related articles.

### 3.2. Most Studied Research Areas and Funding Sources

The most studied research areas in anthrax vaccine were immunology (*n* = 495, 45.41%), research experimental medicine (*n* = 224, 20.55%), infectious diseases (*n* = 215, 19.72%), microbiology (*n* = 198, 18.17%), and biochemistry molecular biology (*n* = 143, 13.12%). Figure 3 shows the most studied research areas in anthrax vaccine between 1991 and 2021.

The majority of the included studies were funded by the USA sources. The sources that funded the highest number of studies were the United States Department of Health and Human Services (HHS) (*n* = 355, 32.57%), the National Institutes of Health (NIH) (*n* = 336, 30.83%), National Institute of Allergy Infectious Diseases (NIAID (*n* = 280, 25.69%), National Institute of General Medical Sciences (NIGMS) (*n* = 37, 3.39%), and the United States Department of Defense (*n* = 32, 2.94%). Table 5 shows the 10 leading funding sources in anthrax vaccine research between 1991 and 2021.

### 3.3. Affiliated Institutions and Countries in Anthrax Vaccine Research

A total of 1151 institutions from 64 countries or regions participated in the included articles. The leading institutions in anthrax vaccine research were the United States Army Medical Research Institute of Infectious Diseases (*n* = 87, citations = 5843), followed by CDC (*n* = 66, citations = 2057), and NIAID, USA (*n* = 37, citations = 1381). Table 6 shows the leading institutions in anthrax vaccine research. Furthermore, the highest collaboration among the leading institutions was between CDC and Emory University, as shown in Figure 4.

The highest contributing countries to anthrax vaccine research are the USA with (*n* = 663, 60.82%) published articles, followed by the UK (*n* = 97, 8.90%), then India (*n* = 64, 5.87%). There were 34 published articles with unknown countries of origin. The most globally cited countries or regions are the USA (*n* = 22,200), followed by the UK (*n* = 3052), then Israel (*n* = 1807). Table 7 shows details of the top 10 countries involved in publishing anthrax vaccine research and Appendix A shows details of all countries involved.

### 3.4. Co-Authorship Countries Overlay Visualization Mapping

The obtained dataset was plotted for co-authorship countries’ overlay visualization mapping. The countries involved in anthrax vaccine research were plotted based on total link strength (TLS), and those with zero TLS were excluded. In Figure 5, the line represents the collaboration between two countries, while the node represents the country’s contribution. The thicker the line, the strongest the collaboration, while the bigger the node, the higher the contribution. The USA, England, Germany, France, and Canada had the highest TLS. In recent years (2020–2021), some developing countries such as Mozambique and Jordan contributed to anthrax vaccine research (Figure 5). Furthermore, Figure 6 shows (a) inter-countries collaboration in anthrax vaccine research, and (b) USA collaboration with other countries. The blue color in the figure shows that, by far, the USA is the largest contributor, and Figure 6b shows that the USA collaborated with every country involved in anthrax vaccine research. However, the USA had the strongest collaboration with the UK, China, Canada, Germany, France, India, Italy, Japan, Turkey, and Australia.

### 3.5. Keywords Analysis and Visualization Mapping

The minimum number of occurrences of a keyword and the minimum cluster size was fixed at 5. Of the total 1678 author keywords used in anthrax vaccine research, only 72 met the criteria. The 20 most frequently used author keywords were plotted forming two clusters, each color in Figure 7a represented a different cluster. The most frequently used author keywords were anthrax (*n* = 232), *Bacillus anthracis* (*n* = 159), vaccine (*n* = 124), protective antigen (*n* = 91), and anthrax vaccine (*n* = 60). The top 20 most appeared keywords from the total of 2319 in KeyWords Plus were plotted into three clusters, each color in Figure 7b represented a different cluster. The most frequently used Keywords Plus were *Bacillus anthracis* (*n* = 293), protective antigen (*n* = 197), vaccine (*n* = 167), toxin (*n* = 165), and immunogenicity (*n* = 163).

### 3.6. Bibliographic Coupling Sources

The obtained dataset was also plotted for bibliographic coupling sources based on the number of citations. The maximum number of documents of a source was selected at 5, while the minimum number of citations of a course was fixed at 50. Of the total 334 sources (journals), 38 sources met the threshold. The top 20 sources based on citations were plotted into two clusters. In Figure 8, the leading bibliographic coupling sources were *Vaccine* (documents = 180, citations = 5695, TLS = 47,142), followed by *Infection and Immunity* (documents = 103, citations = 5533, TLS = 43,013), and *Proceedings of the National Academy of the United States of America* (documents = 22, citations = 2008, TLS = 6429).

### 3.7. Top Authors, Keywords, and Countries Three-Field Plot

The relation between authors, keywords, and countries was mapped using the three-fields plot as shown in Figure 9. The height of the rectangles in the three-field plot originates based on the rate or value of the summation of the relations arising between the plotted components of the three-field plot. The more relations between the plotted components, the higher the rectangle. As shown in Figure 9, the authors mainly engaged with studies using anthrax, *Bacillus anthracis*, vaccine, protective antigen, and anthrax vaccine keywords, and these studies have mainly originated from the USA, the UK, India, Korea, and China.

### 3.8. Thematic Mapping

The thematic map was generated based on author keywords and was plotted into four themes; niche themes (left top), motor themes (top right), emerging or declining themes (left bottom), and basic themes (right bottom), as shown in Figure 10. The centrality on the X-axis shows the degree of interaction of a cluster in comparison with other clusters, while the density on the Y-axis shows the internal strength of a cluster. The motor themes are well developed and important for the structuring of the research field. In the motor themes, mainly studied domains are anthrax, vaccine(s), protective antigen, lethal factor, ava, vaccine safety, recombinant protective antigen, stability, and anthrose. The highly focused domains in niche themes are B. anthracis, spore, protective antigen, neutralization, phagocytosis, macrophage, mucosal vaccine, and toxin. The domains of concern in basic themes are *Bacillus anthracis*, DNA vaccination, mice, genotyping, and virulence. The emerging or decline themes are mainly focused on vaccination, adjuvant(s), anthrax vaccines, anthrax toxin, phage display, bacteriophage, DNA vaccine, immunogenicity, and immune response.

## 4. Discussion

To the best of our knowledge, this is the first bibliometric and visualized study on anthrax vaccine research indexed in the WoSCC database. Bibliometric studies are important and provide comprehensive historical information on scientific publications [41]. Furthermore, bibliometric analyses provide indications of the productivity of countries, authors, and organizations and analyze the structure of publications in a particular research area [42].

The publications’ growth and research trends can reflect the development, advances, and achievements in a specific research area. For example, anthrax vaccine research has increased in the past few decades. However, a small number of studies have been published from 1904 to 1990, while a significant increase has been observed after 2000. Across the globe, anthrax has been used as a weapon for nearly a century [43]. The rise in anthrax vaccine research after 2000 might be due to the mailed powdered anthrax letters in the USA in 2001 [43]. Furthermore, this increase is consistent with the finding of a bibliometric analysis of systematic reviews on vaccines and immunization in general, which found that the number of vaccine-related systematic reviews published annually increased by over 9 times from 2008 to 2016 [44].

In anthrax vaccine research, the USA led the field and significantly produced the highest number of publications compared to other countries. The finding of our study is in line with other studies conducted in different research areas, including tuberculosis [45], myocardial infarction [46], vaccine [47], exosomes [48], rabies vaccine [49], fascioliasis [50], and hepatitis A [36]. Moreover, in terms of citations, and corresponding author country, the USA was the most dominated country. The majority of the institutions affiliated with anthrax vaccine research were from the USA, and nine out of the top ten funding sources are from the USA too. This indicates that the USA allocated a considerable budget to research and development almost in every research area. However, it must be noted that the USA is larger than other developed countries involved in anthrax vaccine research such as the UK. The USA produced 663 anthrax vaccine articles from 1991 until 2021 and the UK produced 97 articles, the population of the USA is over 334 million and the UK is 67 million. If we standardize the article distribution across the population, the USA will have 2 articles per a million population and the UK will have 1.5 articles per million population, indicating the USA will still be leading regardless of the population size. This is consistent with the findings of a bibliometric analysis of *Bacillus anthracis* research that identifies the USA as the leading in terms of publishing, coauthoring, and funding *Bacillus anthracis* research [51]. Overall, it is evident that the developed countries are making the most significant contribution and achievements to research on the anthrax vaccine. Thus, the researchers in the disease burdened countries must seek to collaborate with researchers from the leading countries such as the USA and the UK.

The top two journals (*Vaccine* and *Infection and Immunity*) published nearly 26% of the anthrax vaccine articles. Both journals’ Journal Citation Reports Quartile ranking is Q3 and first published in 1997. In the year 2021, the maximum number of articles were published in *Vaccine* and *PLOS One*. *Vaccine* was a single source that published the maximum number of articles per year in 2007 and 2006. This indicates that the ranking of the journal and the IF is not the driver that attracts authors to publish in journals but the specialty and scope of the journal, especially in areas such as the anthrax vaccine.

The keywords analysis shows that anthrax, *Bacillus anthracis*, vaccine(s), protective antigen, anthrax vaccine(s), vaccination, lethal factor, adjuvant, DNA vaccine, antibody, bioterrorism, immunogenicity, anthrax vaccine adsorbed, immunization, and anthrax toxin were the most relevant author’s keywords. In addition, in the year 2021, the most frequently used keywords were protective antigen, *Bacillus anthracis*, immunogenicity, vaccine, and toxin. The most focused trend topics in the last five years (2017–2021) were double blind, epidemiology, disease, lethal, B surface-antigen, toxins, cattle, adjuvant, host, intramuscular injection, B-cells, and affinity.

Anthrax has been reduced globally due to national programs. However, anthrax is most common in agricultural regions of Central and South America, central and southwestern Asia, southern and eastern Europe, sub-Saharan Africa, and the Caribbean. Anthrax in the USA is rare [52]. The US military views anthrax as a potential biological terrorism threat because anthrax spores are so resistant to destruction and easily spread in the air. In several foreign countries, the development of anthrax as a biological has been reported [53]. Therefore, anthrax vaccine research is considered an important research area.

One of the limitations of this study is that it utilizes a single database (WoSCC). Other databases such as PubMed, Scopus, and Google Scholar may alter the publication’s frequency and citation count. However, WoSCC is the most common database for producing bibliometric analysis. Another limitation is that the search was limited to research articles and the English language. This may affect the number trend and citation of papers published in other languages that have also cited documents included in this study.

## 5. Conclusions

These findings are of interest to researchers and policymakers in providing the bench bibliographic and visualization mapping of the anthrax vaccine-related articles. During the past few decades, the number of studies on anthrax vaccines has increased, mainly focusing on immunology. The most productive year was 2006, with more than 20% of the total articles being published in the period 2006–2008. A decline in the number of publications was observed since 2013. The developed countries made significant contributions to anthrax vaccine-related research. The USA strongly dominated the anthrax vaccine research area, followed by other developed countries, and low-income countries have produced a limited number of studies. The USA was the leading country in terms of citations, corresponding author country, leading institutions, and international collaboration. Among the top 10 leading authors, 6 authors are from the USA. The majority of the top leading institutions are also from the USA. About 90% of the total studies were funded by the United States Department of Health and Human Services (HHS), National Institutes of Health (NIH), USA, and the National Institute of Allergy and Infectious Diseases (NIAID), USA. Strong research collaboration should be established among researchers and institutes from developing countries with developed countries to improve the global output of research in this area.

## Figures and Tables

**Figure 1 vaccines-10-01007-f001:**
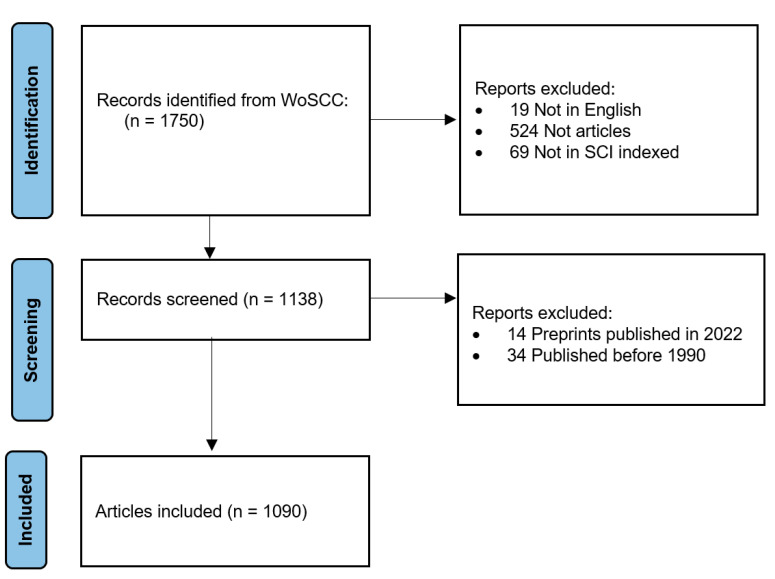
Articles selection flow chart.

**Figure 2 vaccines-10-01007-f002:**
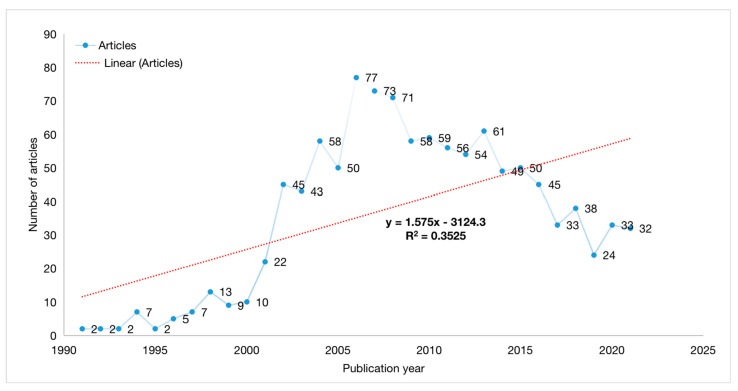
Yearly anthrax vaccine articles published from 1991 to 2021.

**Figure 3 vaccines-10-01007-f003:**
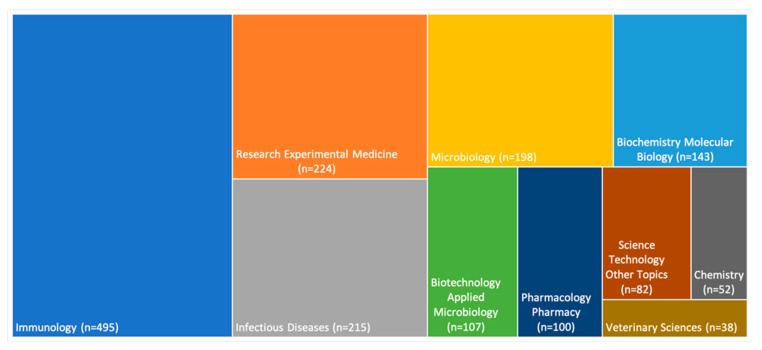
The most studied research areas in anthrax vaccine between 1991 and 2021.

**Figure 4 vaccines-10-01007-f004:**
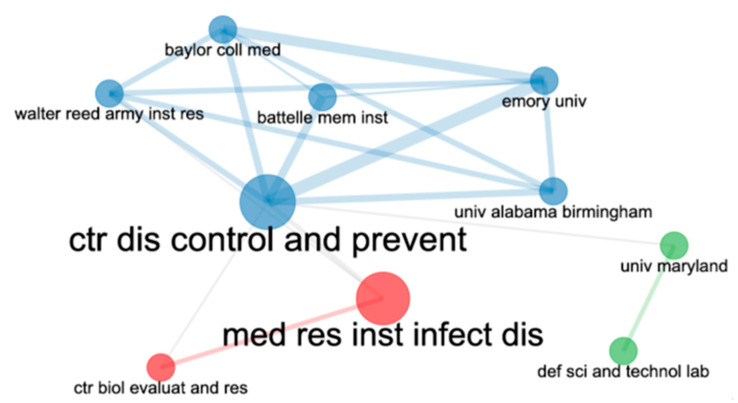
Mapping of the top 10 leading collaborating institutions in anthrax vaccine research between 1991 and 2021.

**Figure 5 vaccines-10-01007-f005:**
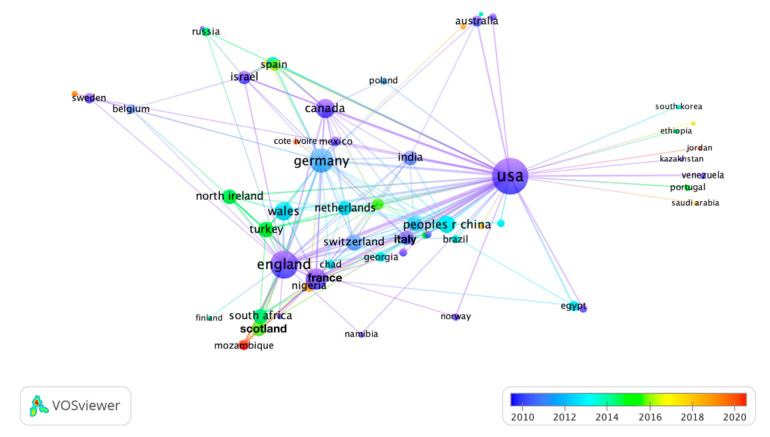
Co-authorship countries overlay visualization mapping over time (year).

**Figure 6 vaccines-10-01007-f006:**
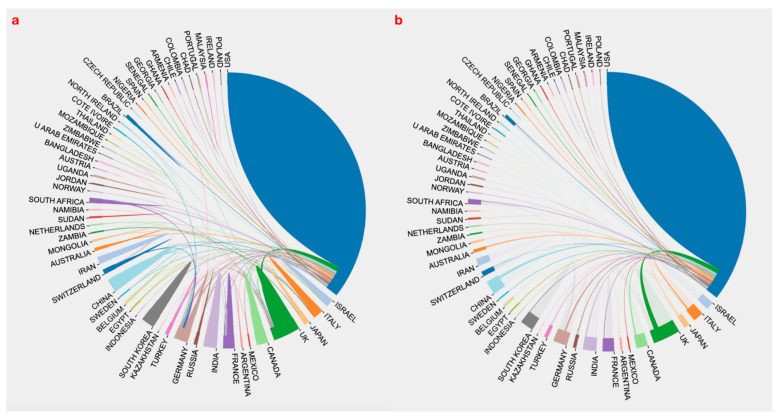
(**a**) Inter-countries collaboration in anthrax vaccine research, (**b**) the USA collaboration with other countries or regions.

**Figure 7 vaccines-10-01007-f007:**
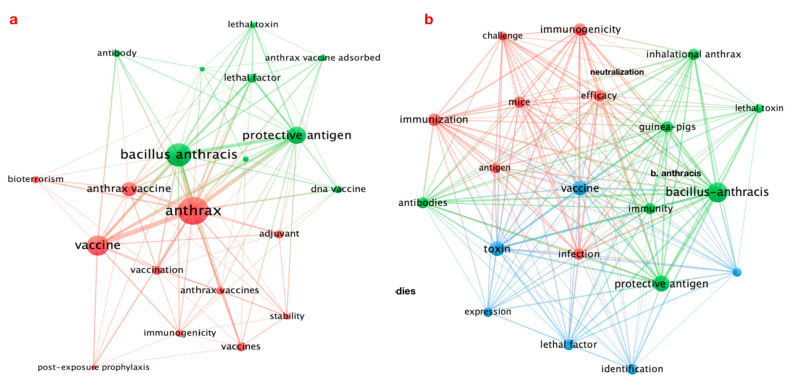
(**a**) Author’s keyword, (**b**) KeyWords Plus network visualization mapping.

**Figure 8 vaccines-10-01007-f008:**
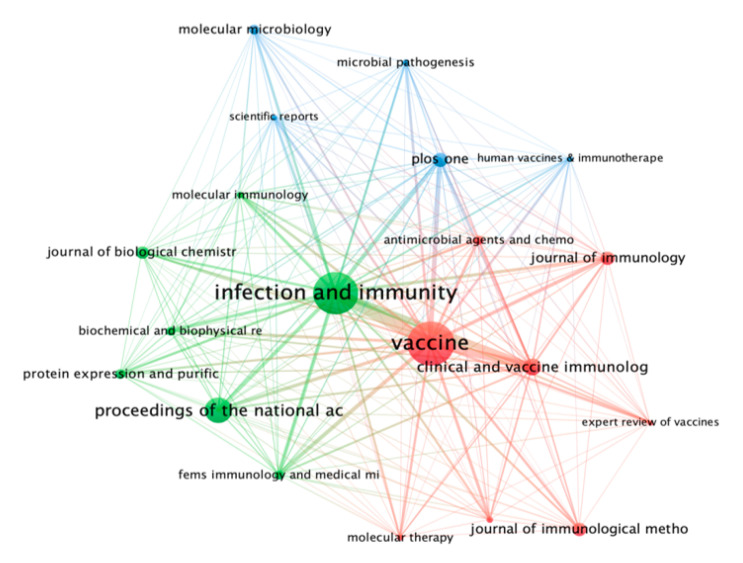
Bibliographic coupling sources network visualization mapping based on the number of citations.

**Figure 9 vaccines-10-01007-f009:**
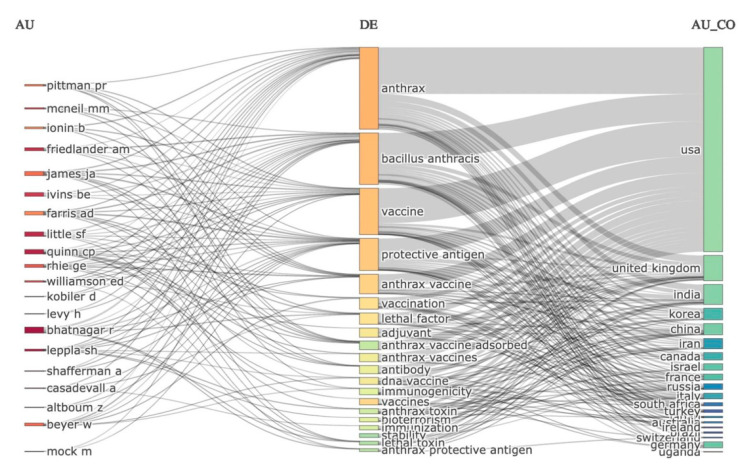
Relations between authors (**left**), keywords (**middle**), and countries (**right**) in anthrax vaccine research.

**Figure 10 vaccines-10-01007-f010:**
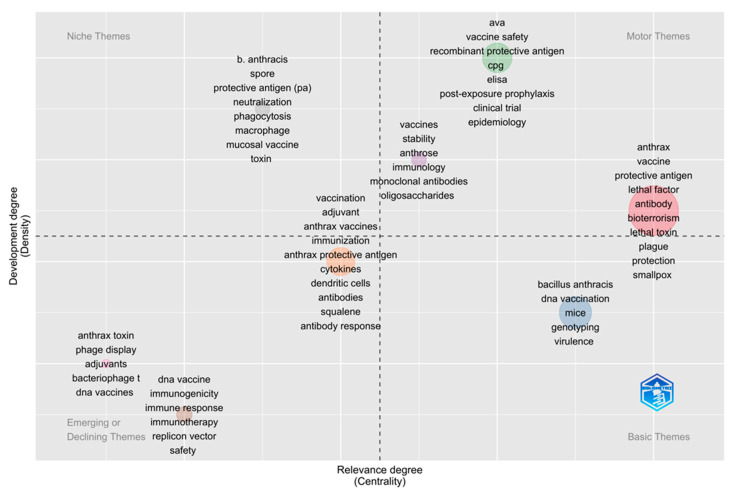
Thematic mapping of anthrax vaccine research.

**Table 1 vaccines-10-01007-t001:** Main description of the included articles.

Description	Results
Time-span	1991–2021
Articles	1090
Journals	334
Institutions	1151
Countries	63
Average publication per year	35.2
Average citations per article	28.75
Average citations per year per article	1.988
Total local citations (within the sample)	6327
Total global citations (within the WoSCC)	31,335
Cited references	41,776
Keywords Plus	2319
Author’s keywords	1678
Authors	4567
Single-authored articles	42
Articles per author	0.239
Authors per article	4.19
Collaboration index	4.32

**Table 2 vaccines-10-01007-t002:** Number published articles and citations in anthrax vaccine per year from 1991 to 2021.

Year	Articles	Citations
2021	32	90
2020	33	78
2019	24	181
2018	38	333
2017	33	302
2016	45	482
2015	50	708
2014	49	798
2013	61	1041
2012	54	1039
2011	56	1271
2010	59	1689
2009	58	1801
2008	71	1929
2007	73	2561
2006	77	2553
2005	50	1959
2004	58	2314
2003	43	2413
2002	45	2388
2001	22	1565
2000	10	590
1999	9	948
1998	13	750
1997	7	423
1996	5	103
1995	2	178
1994	7	328
1993	2	267
1992	2	186
1991	2	67

**Table 3 vaccines-10-01007-t003:** Details of authors who published at least 15 anthrax vaccine-related articles.

Author	Articles (%)	Fractionalized Frequency	Citations (Per Year)	Institution	Country
Rakesh Bhatnagar	35 (3.21%)	8.72	703 (79.45)	Jawaharlal Nehru University	India
Stephen H. Leppla	34 (3.12%)	6.36	1731 (108.60)	NIAID	USA
Conrad P. Quinn	31 (2.84%)	3.43	917 (75.79)	CDC	USA
Stephen F. Little	27 (2.48%)	5.42	1536 (81.78)	United States Department of Defense	USA
Arthur M. Friedlander	24 (2.20%)	3.65	2762 (138.98)	United States Department of Defense	USA
Bruce E. Ivins	19 (1.74%)	3.45	1665 (78.29)	United States Army Medical Research Institute of Infectious Diseases	USA
Michael M. McNeil	19 (1.74%)	2.68	302 (63.10)	CDC	USA
Avigdor Shafferman	18 (1.65%)	2.36	976 (60.69)	Israel Institute for Biological Research	Israel
Ethel D. Williamson	16 (1.47%)	2.25	699 (43.78)	Defense Science and Technology Laboratory, Salisbury	UK
Gi-Eun Rhie	15 (1.38%)	2.22	231 (19.50)	Korea Disease Control and Prevention Agency	Republic of Korea

NIAID: National Institutes of Allergy and Infectious; CDC: Diseases Control and Prevention; USA: United States of America; UK: United Kingdom.

**Table 4 vaccines-10-01007-t004:** Details of journals that published at least 15 anthrax vaccine-related articles.

Journals	Articles (%)	Citations (Per Year)	IF (5-Year)	* Q Ranking	Publisher Address
*Vaccine*	180 (16.51%)	5695 (435.50)	3.641 (3.816)	Q3	Elsevier Sci Ltd., the Boulevard, Langford Lane, Kidlington, Oxford OX 51 GB, Oxon, England
*Infection and Immunity*	103 (9.45%)	5333 (315.80)	3.441 (3.702)	Q3	American Society for Microbiology, 1752 N St NW, Washington, DC 20036-2904
** *Clinical and Vaccine Immunology*	42 (3.85%)	860 (79.70)	4.389 (4.988)	Q2	American Society for Microbiology, 1752 N St NW, Washington, DC 20036-2904
*PLOS One*	38 (3.49%)	607 (70.47)	3.24 (3.788)	Q2	Public Library Science, 1160 Battery Street, STE 100, San Francisco, CA 94111
*PNAS*	22 (2.02%)	2008 (130.96)	11.205 (12.291)	Q1	The National Academy of Sciences, 2101 Constitution Ave NW, Washington, DC 20418

PNAS: *Proceedings of The National Academy of Sciences*; Q: Quartile; IF: Impact Factor. * Q ranking Based on the Journal Citation Reports 2020—Released in June 2021. ** As of January 2018, research in this area is published by ASM’s multi-disciplinary, open access journal *mSphere*. One article was published in *mSphere* and was added to the articles published in *Clinical and Vaccine Immunology*. In 2018, *Clinical and Vaccine Immunology* IF was 3.233.

**Table 5 vaccines-10-01007-t005:** Top 10 leading funding sources in anthrax vaccine research between 1991 and 2021.

Ranking	Funding Sources	Country	Articles (%)
1	United States Department of Health and Human Services (HHS)	USA	355 (32.57%)
2	National Institutes of Health (NIH)	USA	336 (30.87%)
3	National Institute of Allergy Infectious Diseases (NIAID)	USA	280 (25.69%)
4	National Institute of General Medical Sciences (NIGMS)	USA	37 (3.39%)
5	United States Department of Defense (USDOD)	USA	32 (2.94%)
6	National Cancer Institute (NCI)	USA	23 (2.11%)
7	National Institute of Diabetes Digestive Kidney Diseases (NIDDK)	USA	22 (2.02%)
8	National Center for Research Resources (NCRR)	USA	21 (1.93%)
9	National Natural Science Foundation of China (NSFC)	China	20 (1.83%)
10	Defense Threat Reduction Agency (DTRA)	USA	18 (1.65%)

**Table 6 vaccines-10-01007-t006:** The leading institutions in anthrax vaccine research with at least 15 published articles.

	Institutions	Articles	Percentage	Total Global Citations
1	USA Med Res Inst Infect Dis	87	8.0	5843
2	CDC, USA	66	6.1	2057
3	NIAID	37	3.4	1681
4	US FDA	37	3.4	831
5	Jawaharlal Nehru Univ	36	3.3	711
6	Israel Inst Biol Res	31	2.8	1346
7	Def Sci & Technol Lab	25	2.3	768
8	Battelle Mem Inst	23	2.1	391
9	Harvard Univ	22	2.0	875
10	Beijing Inst Biotechnol	21	1.9	190
11	Univ Maryland	21	1.9	1174
12	Emory Univ	20	1.8	916
13	Baylor Coll Med	19	1.7	410
14	Univ Michigan	19	1.7	1088
15	USN	19	1.7	547
16	Emergent BioSolut Inc	16	1.5	248
17	Univ Alabama Birmingham	16	1.5	311
18	Univ Oklahoma	16	1.5	281
19	Walter Reed Army Inst Res	16	1.5	467
20	Oklahoma Med Res Fdn	15	1.4	243
21	Uniformed Serv Univ Hlth Sci	15	1.4	188

**Table 7 vaccines-10-01007-t007:** Top 10 countries involved in anthrax vaccine research between 1991 and 2021.

Rank	Country	Articles	Percentage	Total Global Citations
1	USA	663	60.8	22,200
2	UK	97	8.9	3052
3	India	64	5.9	1070
4	China	55	5.0	737
5	Germany	41	3.8	1044
6	Israel	41	3.8	1807
7	France	36	3.3	1366
8	Unknown	34	3.1	531
9	Canada	27	2.5	687
10	Italy	24	2.2	498

## Data Availability

Not applicable.

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
