# Peer review of "Bibliometric Analysis and Visualization Mapping of Anthrax Vaccine Publications from 1991 through 2021"

_vaccines, 2022, doi:10.3390/vaccines10071007_

Round 1
Reviewer 1 Report
Ahmad et al submitted a manuscript on current research trends on anthrax vaccine. Or so I thought based on the title.
However, in no way the manuscript summarizes global research trends, all the authors do is to summarize:
1) How many papers were published on the topic of anthrax vaccine (in the broadest sense) in total and peer year?
2) How often had the research been cited?
3) Who (which author) published the most papers on the topic?
4) What journals published the most papers on the topic?
5) What was the most studied research area (in the broadest sense)?
6) What were the most used key words?
Unfortunately, this manuscript does not advance the field in any way. This really is a shame because I liked the introduction that (to) briefly summarized several vaccine preparations and clinical trials. The authors may want to consider refocusing and reviewing the actual current research on anthrax. The recent manuscript is in my opinion not publishable.
Author Response
The authors would like to thank the reviewer for taking the time to review our manuscript and for the comments made. The points that the reviewer mentioned in his comment are what is expected to be done in a “Bibliometric Analysis” to summarise the global trend. We would like to highlight that this is not a literature review, and the authors would hope that the reviewer makes the scientific judgment based on the method and scope of the manuscript. The authors have made significant amendments to the manuscript including new analysis, reproduction of the figures, producing new figures and new tables, rewriting the whole abstract, results, and conclusion sections, and amending all other sections. All amendments to the manuscript are made in blue font to facilitate your review.

Reviewer 2 Report
This is a very well-designed comprehensive review analysis type study focusing on research carried out on anthrax vaccine from 1904 to 2021 with special emphasis on global research trends and network visualization mapping. Lots of data were collected and analyzed logically using robust analysis packages. Key findings are presented in a very well manner.
Comments:
Minor comments:
In the introduction, pleaser provides a brief history of B. anthracis and the disease.
Figure. Please add the name of the country those authors belong to, next to their name.
Please increase the quality of Fig. 6, same for Figure 8.
English needs to be improved.
Major comment:
Need to identify and highlight the knowledge gap in relation to Anthrax vaccine and areas that need to be addressed for the eradication of anthrax effectively.
Author Response
The authors would like to thank the reviewer for allowing us to revise our manuscript. The authors also would like to thank the reviewer for the comments and recommendations that we believe have improved the quality of the manuscript. The authors have made significant amendments to the manuscript including new analysis, reproduction of the figures, producing new figures and new tables, rewriting the whole abstract, results, and conclusion sections, and amending all other sections. All amendments to the manuscript are made in blue font to facilitate your review.

Reviewer 3 Report
- The abstract section
· This section should be rewritten:
· What is the exact aim of this paper? I think that authors should solve a problem through this paper with a critical point of view
· Conclusion, is the weakest part of this section
· The same idea is repeated throughout the section
· What is the mean of this sentence at the end of abstract “
-Introduction section
- L50 italic form of Bacillus anthracis throughout the MS
- L58-59, develops more this sentence and use recent and appropriate references
-please clarify the link between Bacillus anthracis and anthrax with an appropriate schema
- Why authors present the history of vaccines against anthrax, how about the mechanism of action
-Results
-authors should develop the Figure 1, criteria and equation, ….
- can authors compare these results with another’s reports?
- the sub section “3.1. Most prolific authors and journals should be deleted!
- in the se subsection 3.2, authors should link all obtained results (Most studied research areas and leading countries)
- fig 4 and 5 are statistically wrong
- what is the objective to this sub section 3.3. Top 10 most cited articles
the overall objective is not at all clear
authors shows numerous papers with authors without a good link
-Discussion
Authors used only one reference!
Poor discussion
Conclusion
Should be improved
-
Author Response

(The authors gave the same response as above.)

Round 2
Reviewer 3 Report
accept as it is. The MS was improved
Author Response
Many thanks